# Advancing Multi-Organ and Pan-Cancer Segmentation in Abdominal CT Scans through Scale-Aware and Self-Attentive Modulation

Pengju Lyu[1,2*][0009−0004−0863−8110], Junchen Xiong[2*][0009−0000−1988−1184], Wei Fang[2][0009−0004−2291−6068], Weifeng Zhang[2][0000−0001−6408−6489], Cheng Wang[2†][0000−0003−1138−337X], and Jianjun Zhu[2,3†][0000−0001−5895−7663]

[1] City University of Macau, Macau, China
[2] Hanglok-Tech Co., Ltd., Hengqin 519000, China
[3] Zhongda Hospital, Medical School, Southeast University, Nanjing 210009, China
{pj.lv, cheng.wang, jj.zhu}@hanglok-tech.cn

**Abstract.** Accurately segmenting abdominal organs and tumors within computed tomography (CT) scans holds paramount significance for facilitating computer-aided diagnosis and devising treatment plans. However, inherent challenges such as lesion heterogeneity and the scarcity of adequately annotated data hamper model development. In this study, we present a two-phase cascaded framework to address the complexities of multi-organ and pan-cancer segmentation. A lightweight CNN first generates candidate regions of interest (ROIs) followed by a hybrid CNN-Transformer model culminating in refined segmentation by synergizing scale-aware modulation for local features and self-attention for global context. Our proposed method secured the 5th position in the MICCAI FLARE23 final test set, showcasing its competitive edge in achieving precise target segmentation with mean Dice Similarity Coefficients of 90.51% for multi-organ and 53.04% for pan-cancer respectively. Additionally, efficient inference is exhibited with an average runtime of 18 seconds per 512 × 512 × 215 3D volume with less than 2G GPU memory consumption. Our code is available at: https://github.com/lyupengju/Flare23.

**Keywords:** Multi-organ and pan-cancer segmentation · Hybrid CNN-Transformer model · Scale-aware and self-attention modulation.

## 1 Introduction

Medical image segmentation plays a crucial role in clinical diagnosis. Accurate organ and cancer segmentation in abdomen computed tomography (CT) as one of the most commonly used modalities for the abdominal diagnosis can assist clinicians in identifying distinct anatomical regions as well as assessing the structure of lesions which assumes critical significance in computer-aided diagnosis,

---

* Equal contribution.
† Corresponding authors.

treatment planning, and image-guided interventions. For instance, the efficacy of radiotherapy treatment planning (RTP), to a great extent, hinges upon the precise demarcation of both the organ at risk (OAR) and the target tumor [45]. Moreover, segmentation on pan-cancer enables the identification of common features and patterns across different cancer types, facilitating the development of targeted therapies and personalized medicine approaches, e.g., identification of unique gene expression signatures associated with different cancers are valuable as diagnostic biomarkers and therapeutic targets [18].

In the deep learning era, the application of convolutional neural network (CNN) or Transformer-based U-Net represents a seminal milestone in the field of medical image segmentation. By virtue of its expansive encoder-decoder structure, U-Net [30] effectively captures both local and global contextual information, enabling the precise delineation of anatomical structures. Its hierarchical approach, coupled with skip connections, facilitates the fusion of multi-scale features, empowering U-Net to discern fine-grained details and accurately segment complex structures in medical images. CNN-based U-Net variants [26] [11] leverage the power of convolutional layers to extract spatial features, enabling the network to discern intricate patterns and variations in tumor morphology, with remarkable precision. On the other hand, Transformer-based U-Net models [7] [32] [44] exploit self-attention mechanisms to capture long-range dependencies and contextual relationships, facilitating a comprehensive understanding of anatomical structures. The hybridization of CNN and Transformer [4][39]stands to the pursuit of synthesizing the best of both paradigms, aiming to forge a sophisticated framework that pushes the boundaries of segmentation accuracy and efficiency.

Abdominal multi-organ and pan-cancer segmentation, however, continues to pose several challenges due to the inherent complexity and variability of cancer lesions, e.g., inter- and intra-tumor heterogeneity coupled with the presence of surrounding anatomical structures that can confound accurate segmentation [25]. On top of that, the scarcity of cancer annotated datasets, especially for rare cancer types, poses a significant hurdle in training accurate and generalizable models. MICCAI FLARE23[§] (Fast, Low-resource, and Accurate oRgan and Pan-cancer sEgmentation in Abdomen CT) makes a significant contribution with the availability of an extensive partial labeled dataset, enabling comprehensive research and analysis in the field. To mitigate the requirement for fully labeled data, which aligns with FLARE23 challenge's objectives, self-training with pseudo labeling and semi-supervised learning emerge as a valuable strategy [21]. Self-training entails the generation of surrogate labels through models trained on partially labelled datasets, thereby offering a bridge towards the realm of fully supervised methodologies. Lian et al. [19] introduces a novel approach that employs partially labelled single-organ datasets to generate pseudo labels for multi-organ segmentation, utilizing partial and mutual priors to enhance organ segmentation performance. Though iterative pseudo labeling with one resource-intensive nnU-Net and selecting reliable ones, Huang et al. [12], un-

---

[§] https://codalab.lisn.upsaclay.fr/competitions/12239

der this knowledge distillation framework, ultimately attain a lightweight model achieving accuracy and efficiency tradeoff in FLARE22 [24]. Semi-supervised learning leverages unlabeled samples to improve generalization [17] where consistency regularization is a popular approach enforcing invariant predictions under input perturbations [33] [15]. Other than that, Pan et al. [27] adopt adversarial training [43] that focuses on training a generator against a discriminator that tries to differentiate segmented outputs derived from labeled versus unlabeled data to promote outputs distribution convergence. On the other hand, the majority of extant deep learning architectures for medical image segmentation, such as APAU-Net [36], TransBTS [16], albeit achieving impressive precision optimized on high-compute laboratory settings with GPUs, typically manifest immense computational demands and parametric complexity. While in bed-side setting with on-device processing of limited computational resources and memory capacities., e.g., point-of-care imaging [34] or interventional surgeries demanding immediate decision-making [45], developing light-weighted, yet competent and scalable models for robust and reliable segmentation becomes paramount.

In this work, we aim to develop a fast, low-resource, and accurate organ and pan-cancer segmentation framework. Our approach is based on the classic two-phase (location-segmentation) cascaded processing stream wherein a lightweight CNN in phase one employing partial convolution and a novel hybrid CNN-Transformer model with synergistic amalgamation of scale-aware modulator and self-attention in phase two are proposed. We trained both models with forementioned simple self-training with pseudo labeling technique. The obtained results on validation set not only demonstrate superior performance on Dice Similarity Coefficient (DSC) and Normalized Surface Dice (NSD) but also showcase favorable inference speeds, underscoring the efficacy and practicality of our proposed method.

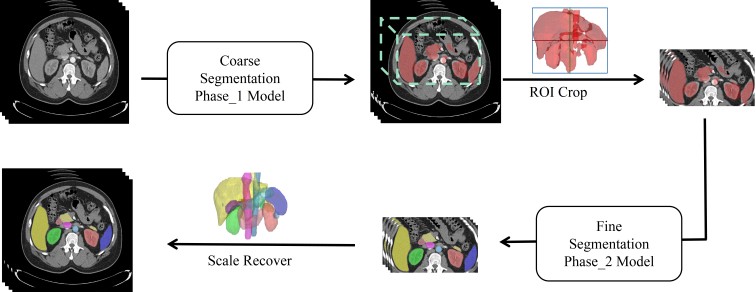

**Fig. 1.** An overview of the two-phase cascade network.

## 2   Method

We adopted localization and segmentation strategy to instantiate multi-phase cascade methodologies which has been proven useful in the past FLARE challenges [36] [35], the overall framework is as shown in the Figure 1. The first phase of the network bestows invaluable location information, furnishing a candidate frame that subsequently facilitates the precise cropping of the image's region of interest (ROI). This localized region (i.e., hard attention [14] [42]), thus extracted, serves as the input for the second-stage network, wherein the process of fine segmentation ensues. This sequential strategy imparts the profound advantage of confining the segmentation focus solely to the target organ, effectively excluding any perturbations arising from unrelated organs or background noise.

### 2.1   Preprocessing

This preprocessing workflow commences with a percentile-based rescaling (percentile values: 5th and 95th) constraining intensity range to crop region containing salient features while suppressing outliers. It is followed by respacing to (1.5mm, 1.5mm, 2mm) rectifies inter-slice spacing disparities, imparting uniformity to the image domain. Image intensities are further Z-normalized to ameliorate convergence dynamics and numerical stability during model training. For phase one, we resize the image dimension to the (128, 128, 128), while patch-wise training method are found to be optimal in identifying tumor, thus four cubes of size (96, 96, 96) are randomly cropped with the ratio between foreground and background equals to 3 : 1. This process culminates in data augmentation where each patch is subjected to random operations, including flipping, rotation, affine, intensity shifting (offset: 0.1), and scaling (scaling factor: 0.1).

### 2.2   Proposed Method

The selection of a lightweight and computationally efficient model is of paramount importance in this framework. The careful choice of the model strikes a delicate balance between computational resource utilization and precision.

**Hierarchical Encoder** We choose to build our model for each phase upon the macro design of U-Net [30] architecture that incorporates multiple levels of hierarchy to capture and process features at different scales as shown in Figure 2. The encoder structure shares across phases with minor variance that stem (patch embedding) block in phase one contains a convolution of kernel size and stride of 4, the number of which halves in phase two. With input size $H \times W \times D$ representing height, width, depth, stem module down scales feature size to corresponding $h \times w \times d$. Base channel number is set as 32 / 60 for each phase respectively at initial stage, which progressive doubles and feature map size $\frac{h}{2^{i-1}} \times \frac{w}{2^{i-1}} \times \frac{d}{2^{i-1}}$, $i \in \{1, 2, 3, 4\}$ reduces itself down the four encoder stages. Between two consecutive stages, down sampling operations is carried out for resolution reduction and channel expansion by a $2 \times 2 \times 2$ convolution with stride 2 followed by layer normalization.

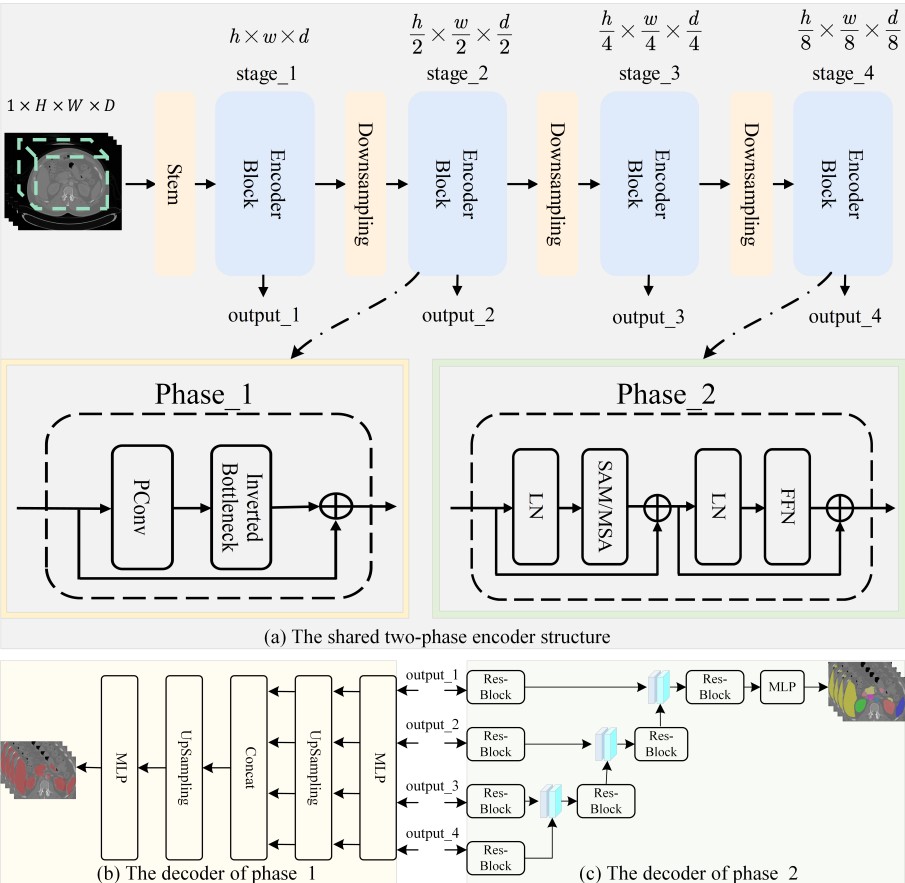

**Fig. 2.** The schematic illustration of proposed models. (a) The shared two-phase backbone structure with Phase_1 model residual inverted bottleneck block where partial convolution (PConv) efficiently conduct spatial token mixing while Phase_2 model utilizing scale-aware modulator (SAM) or Multi-head Self-Attention (MSA) in Metaformer structure; (b) Phase_1 decoder adapted from [38]; (c) Phase_2 decoder adapted from [7].

**Phase_1 Model Components** The localization network is represented by a binary segmentation U-Net, which is designed to treat all labeled organs as the foreground label. To obtain a coarse ROI, we resort to partial convolution (PConv) [2] as choice of spatial token mixing. PConv improves the efficiency by applying filters on only a subset of input channels (first quarter in our case) while preserving the remaining ones. This reduces computational redundancy and the number of memory accesses, resulting in lower FLOPs than regular convolution and higher FLOPS than depthwise convolution. With the completion of shortcut connection and two successive pointwise convolutions, the Phase_1 encoder presents itself as a stacking of residual inverted bottleneck blocks in which channel expansion ratio is 2 and the number of such block is set 2 per stage.

For the decoding of this phase_1, we employ the streamlined MLP decoder from Segformer [38] for efficient information aggregation. Specifically, the multi-level features derived from encoder blocks undergo channel wise compression to base channel number via MLP layers before being upsampled to the size of $h \times w \times d$ and a second MLP layer condenses the concatenated features channels to the number equivalent to that of output classes, trilinear interpolation is ultimately applied to recover to full image size.

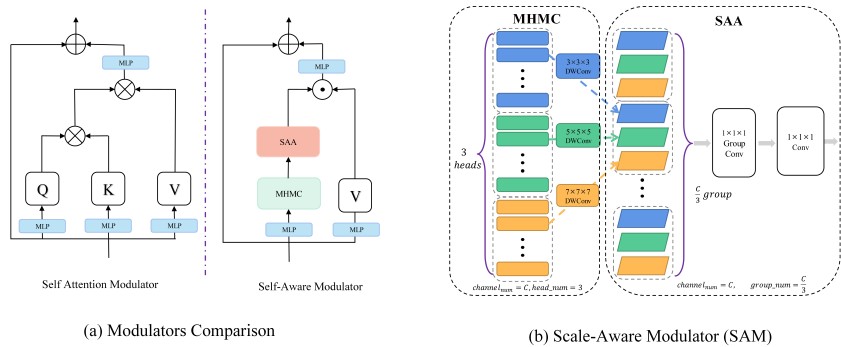

(a) Modulators Comparison

(b) Scale-Aware Modulator (SAM)

**Fig. 3.** (a) Spatial modulation comparison between self-attentive and scale-aware operator. self-attention first generates the key K, query Q, and value V using MLP layers and the weights to modulate the V representations are determined by attention weights computed by measuring the similarity between Q and K. SAM instead directly obtain the weights with Multi-Head Mixed Convolution (MHMC) and a Scale-Aware Aggregation (SAA) blocks. (b) Evolving from [20], the schematic illustration of SAM integrating multi-scale contexts via a MHMC and adapts token representations through a SAA.

**Phase_2 Model Components** For fine segmentation, by taking advantage of the strengths of both CNNs and Transformers in Meta-former style [40], which contains a spatial token mixing layer and a feed-forward layer (FFN) [29]. We

adopt Scale-Aware Modulator (SAM) [20] to reweight the value representations for lower-level local feature extraction in early stages while Multi-head Self-Attention (MSA) [7] dedicated to global information in later stages, see Figure 3 for details. SAM consists of a Multi-Head Mixed Convolution (MHMC) and a Scale-Aware Aggregation (SAA) module to enable the integration of multi-scale contexts and adaptive modulation of tokens. Together, SAM and MSA provide complementary modeling of multi-scale local features and long-range global contexts. Their combination enables extracting both localized fine details and overall spatial relationships.

The MHMC introduces multiple depth-wise convolutions with different kernel sizes, enabling it to capture various spatial features across multiple scales. Figure 3 illustrates the structure of MHMC, wherein the input channels are divided into multiple groups (heads), each subjected to depth-wise separable convolutions with diverse kernel sizes respectively, which are able to discern a diverse spectrum of granularity features in an adaptive fashion.

The SAA module engages in a practice of cross-group information aggregation across all features to harmonize diverse insights from distinct groups. Specifically, three mixed groups are curated with each selecting one channel from previously partitioned group, and the inverse bottleneck structure (expansion ratio $= 2$) with point-wise convolutions are subsequently leveraged fostering a holistic synergy of knowledge propagation and enriched representation, which, by means of the Hadamard product operation, eventually serves as weight modulator of the value V in contrast to yielding attention matrices via a matrix multiplication between the query and key in self-attention. The whole process of SAM can be summarized in the following steps:

$$
\begin{aligned}
&\boldsymbol{Input}: \ \boldsymbol{z} \in R^{C \times H' \times W' \times D'} \\
&\boldsymbol{MHMC}: \ H_j^i = DWConv_{k_j \times k_j \times k_j}\left(\boldsymbol{z}_j^i\right), j \in \{1, 2, \cdots, M\}, i \in \{1, 2, \cdots, C/M\} \\
&\boldsymbol{SAA}: \ G_i = Relu\left(IN\left(Conv_{1 \times 1 \times 1}\left(\left[H_1^i, H_2^i, \cdots, H_M^i\right]\right)\right)\right) \\
&\qquad\quad W = Conv_{1 \times 1 \times 1}\left(\left[G_1, G_2, \cdots, G_{C/M}\right]\right) \\
&\boldsymbol{Output}: \ \hat{\boldsymbol{z}} \ = \ W \odot \left(Conv_{1 \times 1 \times 1}\left(\boldsymbol{z}\right)\right)
\end{aligned}
\tag{1}
$$

Let $\boldsymbol{z} \in R^{C \times H' \times W' \times D'}$ denote the input tensor to the SAM module with $C$ channels and spatial dimensions $H' \times W' \times D'$ for the current layer. We divide the channels into $M = 3$ heads, indexed by $j \in \{1, 2, 3\}$, with $C/M$ channels in each head. The output is denoted as $\hat{\boldsymbol{z}}$ with the same dimensions as $\boldsymbol{z}$. Within each head j, we have single-channel feature maps $\boldsymbol{z}_j^i \in R^{1 \times H' \times W' \times D'}$ for $i \in \{1, 2, \cdots, C/M\}$. These are convolved with learned depth-wise kernels $DWConv$ of size $k_j$, where we set $k_j \in \{3, 5, 7\}$ for the 3 heads respectively. $\odot$ denotes dot product operation.

SAM blocks reside only in the initial two stages. During the penultimate stage, triple of SAM blocks and Multi-Head Self-Attention (MSA) blocks are alternatively stacked, effectively capturing the transition from local to global dependencies. In the ultimate stage, exclusively MSA blocks are employed, thereby

ensuring proficient capture of long-range dependencies. The number of such blocks in each stage amounts to $2, 4, 6, 2$ correspondingly.

We adopted phase_2 decoder similar to that from UNETR [7]. A residual block, composed of two consecutive sequences of Conv + InstanceNorm + LeakyRelu, is applied to skip connections as well as subsequent concatenated features. Upsampling is realized with transpose convolution.

### 2.3   Post-processing

After phase one, we remove objects of size smaller than $(20 \times 20 \times 20)$, which might be outliers affecting a precise ROI cropping for phase two whose result are refined by preserving solely the largest components of organs. Based on the observation that predicted tumor mask could appear separate with abdominal organs though within the ROI defined by bounding box. This contradicts a well-established fact that tumors originate on organs. We have tumor mask through basic morphological operations of dilation and subtraction to identify any organs in proximity, thereby filtering out those isolated components as shown in Figure 4. The resultant mask are finally mapped back to the same size of input image.

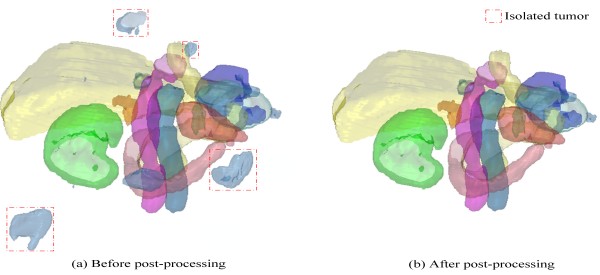

(a) Before post-processing                    (b) After post-processing

**Fig. 4.** Feasibility analysis of post-processing operations. It is evident that the proposed post-processing applied to the predictive mask effectively eliminates isolated tumors.

## 3   Experiments

### 3.1   Dataset

The FLARE23 challenge constitutes an extension of its precursor, the FLARE 2021-2022 initiative [23] [24]. Its primary objective is to foster the advancement of foundational models in the realm of abdominal disease analysis. The delineation objectives encompass a spectrum of 13 distinct organs including liver, spleen, pancreas, right kidney, left kidney, stomach, gallbladder, esophagus, aorta, inferior vena cava, right adrenal gland, left adrenal gland, and duodenum in addition

to diverse abdominal lesions, namely pan-cancer. The training dataset is curated from more than 30 medical centers under the license permission, including TCIA [3], LiTS [1], MSD [31], KiTS [8,9], autoPET [6,5], TotalSegmentator [37], and AbdomenCT-1K [25]. The training dataset consists of a total of 4000 abdominal CT scans in which 2200 scans with partial annotations and 1800 scans devoid of annotations. Two sets of 4000 pseudo labels of multi organs, generated by two top-performance teams during FLARE22 [12] [35], were appended afterwards. The validation and testing sets include 100 and 400 CT scans, respectively, which cover various abdominal cancer types, such as liver cancer, kidney cancer, pancreas cancer, colon cancer, gastric cancer, and so on. The organ annotation process used ITK-SNAP [41], nnU-Net [13], and MedSAM [22].

### 3.2 Implementation details

**Table 1.** Development environments and requirements.

| System | Ubuntu 20.04.5 LTS |
|---|---|
| CPU | Intel(R) Xeon(R) Platinum 8358 CPU @ 2.60GHz |
| RAM | 1.0 Ti; 3200 MT/S |
| GPU (number and type) | Two NVIDIA A800 80G |
| CUDA version | 11.8 |
| Programming language | Python 3.8.16 |
| Deep learning framework | torch 2.0.1, torchvision 0.15.2 |
| Specific dependencies | monai 1.2.0 |
| Code | https://github.com/lyupengju/Flare23 |

Throughout the entire experimental process, we implemented our code based on PyTorch library[¶] and MONAI framework[‖]. All models were trained on two Nvidia A800 GPUs. To accelerate model training, the CacheDataset method in the MONAI was utilized for data pre-loading. During the training phase, the Adam optimizer was adopted with weight decay of$1e^{-5}$ to minimize the most widely used joint loss function, i.e., dice and cross entropy [7]. Initial learning rate was set as $3e^{-4}$ scheduled by cosine annealing strategy. The number of training epochs was up to 300 with batch size of 4. See Table 1, 2 for more training and environment settings.

### 3.3 Training protocols

Leveraging the entire dataset comprising 4000 cases and one set of their corresponding organ pseudo labels from FLARE22 winning algorithm [12], we are able to obtain our Phase_1 model by means of a label filtering technique, along

---

[¶] http://pytorch.org/

[‖] https://monai.io/

**Table 2.** Training protocols.

| Network initialization | Random |
|---|---|
| Batch size | 4 |
| Patch size (Phase_2 model) | $96 \times 96 \times 96$ |
| Resized size (Phase_1 model) | $128 \times 128 \times 128$ |
| Total epochs | 300 |
| Optimizer | AdamW |
| Initial learning(lr) | $3e^{-4}$ |
| Lr decay schedule | Cosine annealing |
| Training time for each model | 36 hours |
| Loss function | Dice loss and Cross entropy loss |
| Number of model parameters (Phase_1 / Phase_2) | 1.38 M / 35.84 M |
| Number of flops (Phase_1 / Phase_2) | 1.56 G / 374.77 G |

with a pre-trained Phase_2 model, the specific process is as depicted in Figure 5. Similar to [12], we adopted self-training with pseudo labeling strategy to obtain final Phase_2 model. Specially, we reassigned pseudo annotations in conjunction with 2200 partial ground truth for the whole dataset to update the segmentation model. This process facilitated the creation of a comprehensive dataset, complete with fully annotated organs and tumors. The process of pseudo labeling was executed iteratively 3 times, thereby enabling the iterative enhancement of the quality of pseudo annotations, which is pivotal in advancing the model's performance. In practice, we first split the renewed dateset into two folds, the updating pseudo labels was then formed by ensembling two branch networks through soft voting, which are later utilized to train our final Phase_2 model. We empirically selected the model that tend to produce oversegmented results on pan-cancer, which generally yield better Dice score on online validation leaderboard.

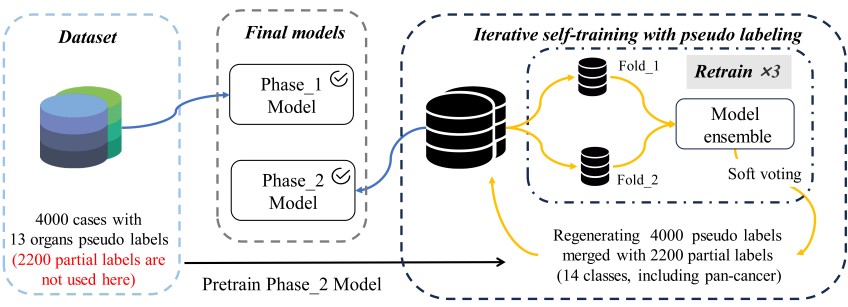

**Fig. 5.** Training pipeline.

## 4    Results and discussion

We conducted comprehensive quantitative evaluation of our proposed model using standard segmentation and efficiency metrics. Regarding accuracy, we report the Dice similarity coefficient (DSC) and normalized surface Dice (NSD) between predicted and ground truth organ and lesion masks with DSC elucidating overall overlap and NSD focusing on boundary alignment precision [32]. Efficiency-wise, running time and the GPU memory consumption, are integral for assessing the algorithm's practicality and real-world applicability. The running time and GPU memory consumption are considered within tolerances of 15 seconds and 4 GB, respectively.

**Table 3.** Quantitative evaluation results in terms of DSC and NSD for organs and tumor respectively.

| Target | Public Validation | | Online Validation | | Testing | |
|---|---|---|---|---|---|---|
| | DSC (%) | NSD (%) | DSC (%) | NSD (%) | DSC (%) | NSD (%) |
| Liver | 97.68 ± 0.54 | 99.28 ± 0.77 | 97.64 | 99.24 | 97.04 | 97.92 |
| Right Kidney | 95.90 ± 2.94 | 96.95 ± 4.27 | 94.86 | 95.95 | 95.54 | 95.42 |
| Spleen | 96.89 ± 1.57 | 98.46 ± 4.13 | 96.19 | 98.06 | 96.54 | 98.47 |
| Pancreas | 85.96 ± 7.20 | 96.67 ± 6.09 | 84.63 | 95.79 | 89.14 | 97.32 |
| Aorta | 94.17 ± 4.39 | 97.34 ± 5.82 | 94.72 | 98.08 | 95.35 | 99.31 |
| Inferior vena cava | 90.16 ± 5.63 | 92.93 ± 5.77 | 89.46 | 91.88 | 90.76 | 93.76 |
| Right adrenal gland | 83.66 ± 1.25 | 95.93 ± 1.39 | 83.97 | 96.58 | 83.67 | 96.24 |
| Left adrenal gland | 84.67 ± 5.47 | 96.73 ± 4.13 | 83.98 | 95.90 | 84.50 | 96.16 |
| Gallbladder | 88.28 ± 19.06 | 90.81 ± 20.10 | 88.92 | 91.09 | 84.84 | 88.15 |
| Esophagus | 80.78 ± 17.86 | 91.23 ± 17.34 | 82.04 | 92.84 | 87.86 | 97.25 |
| Stomach | 94.46 ± 3.09 | 97.75 ± 3.42 | 94.50 | 97.70 | 94.71 | 97.48 |
| Duodenum | 83.07 ± 8.72 | 94.74 ± 6.59 | 83.41 | 94.70 | 86.48 | 95.51 |
| Left kidney | 93.06 ± 14.38 | 94.05 ± 15.32 | 93.60 | 94.80 | 93.41 | 94.40 |
| Organ Average | 89.90 | 95.61 | 89.84 | 95.56 | 90.51 | 95.88 |
| Tumor | 54.25 ± 36.10 | 49.65 ± 33.51 | 50.26 | 45.31 | 53.04 | 44.47 |

### 4.1    Quantitative results

To validate the efficacy of the model, we present in Table 3 the details of 50 cases from the validation dataset, the online validation and the final testing outcomes. Our model demonstrates strong performance on both organ and pan-cancer segmentation from abdominal CTs. For the 13 organs on online validation, we achieve competitive accuracy score with DSC ranging from 82.04% (Esophagus) to 97.64% (liver), and NSD all over 90%, which highlight our model's ability to capture fine anatomical details. Specifically, our model in Phase_2 with only 35.84M parameters achieves considerable gains on average Dice over prior arts spanning CNN-based V-Net (67.70M) [26], nnUNet (30.74M) [13], and Transformer-based Swin UNETR (69.94M) [32], nnFormer (158.9M) [44] as well

as their hybrid CoTr (41.93M) [39], as presented in Figure 6. This again validates the benefits of synergistically combining SAM and MSA from both paradigms.

With regards to pan-cancer segmentation, although our approach attains a relatively high average DSC of 50.26% across all lesion types, since the fact that best model was selected based on its performance on the public 50 cases, the divergence on tumor metrics between it and full validation set coupling with a high standard variance (36.10%) indicates that model's weak capacity of learning generalizable representations of pan-cancer. Our methodology distinguished itself by securing a commendable 5th position in the final test set, quantified by elevated mean Dice on both multi-organ (90.51%) and tumor (53.04%).

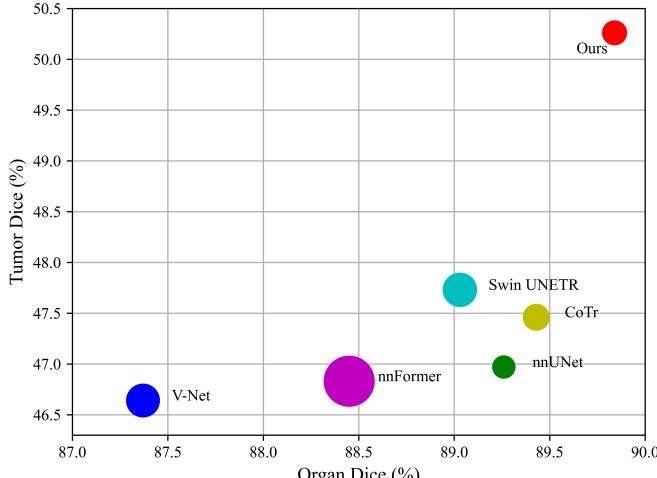

**Fig. 6.** Phase_2 models comparison with prior arts on online validation. The diameter of the circular data points is proportional to the total number of parameters in each respective model.

To analyze the impact of training set size, an ablation study was conducted comparing validation performance between models trained on the full 4000 case dataset versus the 2200 partially labeled cases alone. Despite nearly doubling the training data through pseudo-labeling, the models seem not to learn novel anatomical representations but rather fine-tuning of existing feature spaces, exemplified by both DSC and NSD metrics on either scenario revealing negligible differences regarding organs (0.1%) and tumor (0.5%), as shown in Figure 7, which indicates that model's learned features might not be universally applicable, resulting in limited generalization to different cases, which in turn impacts the overall effectiveness of pseudo labeling.

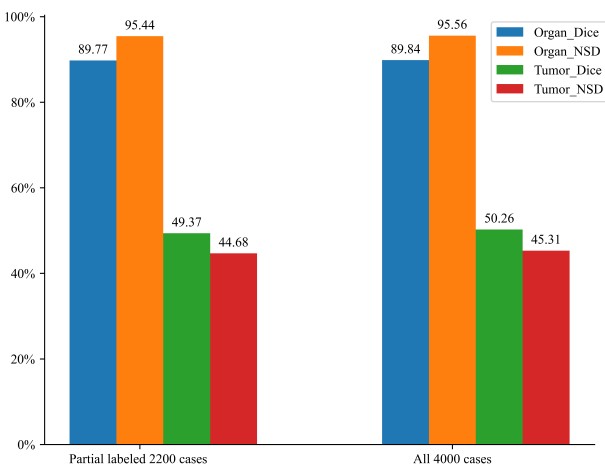

**Fig. 7.** Performance comparison on online validation using 2200 partially labeled examples and 4000 fully labeled examples for training.

## 4.2    Qualitative results on validation set

We supplement our quantitative results with qualitative analysis to gain further insights, as shown in Figure 8. Notably, the segmentation performance exhibits variability across organs. In contrast to near perfect demonstration (Case #27), our model generates fragmentary or inaccurate contours with smaller structures like esophagus and duodenum (Case #69), echoed by their relatively lower Dice scores on validation set. For pan-cancer, while some tumor instances (Case #35) are effectively segmented, showcasing a robust alignment with ground truth annotations, others exhibit violent segmentation inconsistencies (Case #99). This variance in tumor segmentation proficiency is indicative of the complexity inherent in cancer lesions, often characterized by diverse morphological traits and inter-tumor heterogeneity. Column (c) represents the segmentation result by model trained only with partial-label 2200 cases demonstrating similar performance with that of column (d) using all 4000 cases.

## 4.3    Segmentation efficiency results on online validation

Our two-phase cascaded network provides major speed and memory benefits. Table 4 provides the efficiency for certain examples from the validation dataset. For the majority of test cases, our proposed method can complete the inference process requiring extra seconds (8 in average) than the prescribed time budget of 15 seconds, while maintaining GPU memory consumption well under the allotted 4GB limit. Moreover, running time appears to exhibit a positive correlation with input image size owing to the serial scanning nature of the sliding window, traversing spatially across the input, consequently inflicting a computational burden that scales directly with image area, as evidenced by the near 31 second

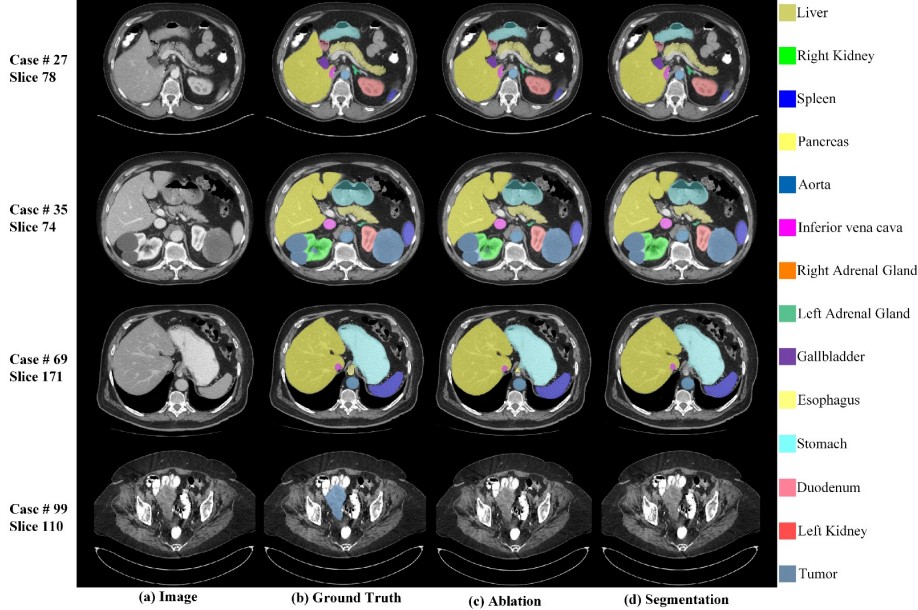

**Fig. 8.** Qualitative evaluation on four cases from validation set.

run time for the largest case 0029 scan, resulting in greater cumulative GPU utilization.

### 4.4   Limitation and future work

While our method shows promise for multi-organ and tumor segmentation, enabling clinical utilization through efficient computation and memory usage. These validation results highlight areas for continued future refinement, especially enhancing delineation of tiny organs and handling greater tumor heterogeneity. For that, tumor synthesis technique [10] could be employed to artificially generate additional lesion examples. This data augmentation approach may facilitate greater robustness in the segmentation model, allowing it to generalize more effectively to the heterogeneity inherent in pathological anatomy. Since our pseudo labeling approach is mostly off-line making impossible real time updating, we should further explore online semi-supervised method as well as mechanisms to enhance the fidelity and reliability of generated pseudo labels such as applying confidence thresholding, and detecting out-of-distribution pseudo labels [21].

## 5   Conclusion

In the pursuit of advancing the state of the art in multi-organ and pan-cancer image segmentation, we have made significant strides in this realm by our par-

**Table 4.** Quantitative evaluation of segmentation efficiency in terms of the run- ning time and GPU memory consumption. Total GPU denotes the area under GPU Memory-Time curve. Evaluation GPU platform: NVIDIA QUADRO RTX5000 (16GB).

| Case ID | Image Size | Running Time (s) | Max GPU (MB) | Total GPU (MB) |
|---------|-----------|-----------------|-------------|---------------|
| 0001 | (512, 512, 55) | 15.17 | 2020 | 17200 |
| 0051 | (512, 512, 100) | 16.73 | 2020 | 22141 |
| 0017 | (512, 512, 150) | 17.84 | 2020 | 23832 |
| 0019 | (512, 512, 215) | 18.01 | 2020 | 23875 |
| 0099 | (512, 512, 334) | 20.61 | 2020 | 27427 |
| 0063 | (512, 512, 448) | 24.40 | 2020 | 32647 |
| 0048 | (512, 512, 499) | 25.36 | 2020 | 33937 |
| 0029 | (512, 512, 554) | 30.87 | 2020 | 43991 |

ticipation in the MICCAI FLARE23 challenge through the development and application of a two-phase cascade framework. Phase_1 model built upon partial convolution enjoys computational efficiency while yielding credible segmented ROI. The harmonious fusion of scale-aware and self-attentive modulation forms the foundation of our Phase_2 model backbone, enabling enhanced segmentation accuracy. Through meticulous model selection, tuning, and optimization, our algorithm has shown promising overall results with reference to precision and efficiency metrics on the online validation and test datasets, substantiating its efficacy in target segmentation. We believe our approach holds the promise of enhancing clinical practices and contributing to the broader scientific understanding of complex medical image analysis in abdominal oncology.

**Acknowledgements** The authors of this paper declare that the segmentation method they implemented for participation in the FLARE 2023 challenge has not used any pre-trained models nor additional datasets other than those provided by the organizers. The proposed solution is fully automatic without any manual intervention. We thank all the data owners for making the CT scans publicly available and CodaLab [28] for hosting the challenge platform.

The study was supported by National Natural Science Foundation of China (81827805, 82130060, 61821002, 92148205), National Key Research and Development Program (2018YFA0704100, 2018YFA0704104). The project was funded by China Postdoctoral Science Foundation (2021M700772), Zhuhai Industry-University-Research Collaboration Program (ZH22017002210011PWC), Jiangsu Provincial Medical Innovation Center (CXZX202219), Collaborative Innovation Center of Radiation Medicine of Jiangsu Higher Education Institutions, and Nanjing Life Health Science and Technology Project (202205045). The funding sources had no role in the writing of the report, or decision to submit the paper for publication.

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
