# OpenReview forum: "Advancing Multi-Organ and Pan-Cancer Segmentation in Abdominal CT Scans through Scale-Aware and Self-Attentive Modulation"
_MICCAI.org/2023/FLARE — Submitted to FLARE 2023_

### Official Review · Reviewer_ABgE · 2023-10-04
**Advancing Multi-Organ and Pan-Cancer Segmentation in Abdominal CT Scans through Scale-Aware and Self-Attentive Modulation**

**Rating:** 7
**Confidence:** 4

**Review:**

The paper proposes a two-stage cascade framework to address the complexity of multi-organ and pan-cancer segmentation. Experiments performed on the MICCAI FLARE23 challenge leaderboard validate the good performance of the method, achieving high segmentation accuracy with average dice similarity coefficients of 89.84 % and 50.26 % for multi-organ and tumor regions, respectively. The paper as a whole is more complete, but still has some grammatical problems.

---

### Official Review · Reviewer_MXAR · 2023-10-04
**Advancing Multi-Organ and Pan-Cancer Segmentation in Abdominal CT Scans through Scale-Aware and Self-Attentive Modulation**

**Rating:** 7
**Confidence:** 4

**Review:**

The paper proposes a two-phase cascaded framework to address the complexities of multi-organ and pan-cancer segmentation. In the first stage is a binary segmentation U-Net. In the second stage, SAM and MSA are combined, which are strong manifestations. The results are impressive with very high organ and tumor DSC and NSD.

---

### Official Review · Reviewer_eGLw · 2023-10-06
**Tow-phase cascaded framework**

**Rating:** 7
**Confidence:** 5

**Review:**

In this paper, the authors aim to address the intricacies of multi-organ and pan-cancer segmentation within computed tomography (CT) scans. The lesion heterogeneity and the lack of comprehensive annotated data are pressing challenges. The proposed two-phase cascaded framework seems innovative. They employ a lightweight CNN to delineate candidate regions of interest (ROIs), which is then succeeded by a hybrid CNN-Transformer model. This model strategically leverages scale-aware modulation for localized features while concurrently harnessing self-attention mechanisms to grasp global contexts. The method has been validated using the MICCAI FLARE23 challenge leaderboard, demonstrating promising segmentation accuracy. The reported Dice similarity coefficients and efficient inference metrics suggest both the robustness and efficiency of their approach. However, the discrepancy between multi-organ and tumor segmentation accuracies warrants further exploration.

---

### Official Review · Reviewer_R6nS · 2023-10-18
**Good paper, but has some issues**

**Rating:** 7
**Confidence:** 4

**Review:**

This study presented a two-phase cascaded framework to address the complexities of multi-organ and pan-cancer segmentation. The proposed method achieved segmentation accuracy with average Dice similarity coefficients of 89.84 % and 50.26 % for multiorgan and tumor regions respectively.
Cons:
1. Isolated tumors were filtered in post-processing, but tumors outside the abdominal region, such as prostate tumors, were present in the whole-body pan-tumor segmentation task. Direct filtering ignores tumors that may be present in other regions.

---

### Decision · Program_Chairs · 2023-10-24

Accept